# Found Graph Data and Planted Vertex Covers

**Austin R. Benson**
Cornell University
arb@cs.cornell.edu

**Jon Kleinberg**
Cornell University
kleinber@cs.cornell.edu

## Abstract

A typical way in which network data is recorded is to measure all interactions involving a specified set of *core nodes*, which produces a graph containing this core together with a potentially larger set of *fringe nodes* that link to the core. Interactions between nodes in the fringe, however, are not present in the resulting graph data. For example, a phone service provider may only record calls in which at least one of the participants is a customer; this can include calls between a customer and a non-customer, but not between pairs of non-customers. Knowledge of which nodes belong to the core is crucial for interpreting the dataset, but this metadata is unavailable in many cases, either because it has been lost due to difficulties in data provenance, or because the network consists of "found data" obtained in settings such as counter-surveillance. This leads to an algorithmic problem of recovering the core set. Since the core is a vertex cover, we essentially have a *planted vertex cover* problem, but with an arbitrary underlying graph. We develop a framework for analyzing this planted vertex cover problem, based on the theory of fixed-parameter tractability, together with algorithms for recovering the core. Our algorithms are fast, simple to implement, and out-perform several baselines based on core-periphery structure on various real-world datasets.

## 1 Partially measured graphs, data provenance, and planted structure

Datasets that take the form of graphs are ubiquitous throughout the sciences [4, 23, 49], but the graph data that we work with is generally incomplete in certain systematic ways [28, 33, 34, 36, 39]. A common type of incompleteness comes from the way in which graph data is generally measured: we observe a set of nodes $C$ and record all the interactions involving this set of nodes. The result is a measured graph $G$ consisting of this *core set* $C$ together with a a potentially larger set of additional *fringe nodes*—the nodes outside of $C$ that interact with some node in $C$. For example, in constructing a social network dataset, we might study the employees of a company and record all of their friendships [54]. From this information, we now have a graph that contains all the employees together with all of their friends, including friends who do not work for the company. This latter group constitutes the set of fringe nodes in the graph. The edge set of the graph $G$ reflects the construction: each edge involves at least one core node, but if two nodes in the fringe have interacted, it is invisible to us and hence not recorded in the data.

E-mail and other communication datasets typically have this core-fringe structure. For example, the widely-studied Enron email graph [24, 37, 43, 55] contains tens of thousands of nodes; however, this graph was constructed from the email inboxes of fewer than 150 employees [35]. The vast majority of the nodes in the graph, therefore, belong to the fringe, and their direct communications are not part of the data. The issue comes up in much larger network datasets as well. For example, a phone service provider has data on calls and messages that its customers make both to each other and to non-customers, but it does not have data on communication between pairs of non-customers. Another example would be a massive online social network may get some information about the contacts of its users with a fringe set comprised of of people not on the system—often including entire countries that do not participate in the platform—but generally not about the interactions taking place in this fringe set. Similarly, Internet measurements at the IP-layer from individual service providers give

only a partial view of the global Internet network [58, 60]. Therefore, graph data often takes the form illustrated in Figure 1a: the nodes are divided into a core and a fringe, and we only see edges that involve a member of the core. Thus, the core set is a *vertex cover* of the underlying graph—since a vertex cover, by definition, is a vertex set incident to all edges.

Often, a graph dataset is annotated with metadata about which nodes belong to the core, and this is crucial for correctly interpreting the data. But there are a number of important contexts where this metadata is unavailable, and we do not know which nodes constitute the core. In other words, at some point, we have "found data" that has core-fringe structure, but the labels identifying the core nodes are missing. One reason for this scenario is that metadata is lost over time for a wide range of reasons: data is repeatedly shared and manipulated, managers of datasets change jobs, and hard drives are decommissioned. This is a central theme in data provenance, lineage, and preservation [11, 44, 56, 59], an especially challenging issue in modern digitization efforts [38] and large-scale data management [32]. A concrete example is the following. Suppose a telecommunications company shares an anonymized dataset of telephone call records with a university but does not include on which nodes were customers and which were the fringe set of non-customers. By the time a student begins to analyze the data and realizes the metadata is missing and important for analysis, the researchers at the company who assembled the dataset have left. At this point, there is no easy way to reconstruct the metadata.

These issues also arise in current research on counter-surveillance [29]. Intelligence agencies may intercept data from adversaries conducting surveillance and build a graph to determine which communications the adversaries were recording. In different settings, activist groups may petition for the release of surveillance data by governments or infer it from other sources [47]. In these cases, the "found data" consists of a communication graph in which an unknown core subset of the nodes was observed, and the remainder of the nodes in the graph (the fringe) are there simply because they communicated with someone in the core. In such situations, there generally is not any annotation to distinguish the core from the fringe. In this case, the core nodes are the compromised ones, and identifying the core from the data can help to warn the vulnerable parties, hide future communications, or disseminate misinformation.

**Planted Vertex Covers.** Here we study the problem of recovering the set of core nodes in found graph data, motivated by the range of settings above. We can view this as a *planted vertex cover problem*: we are given a graph $G$ in which an adversary knows a specific vertex cover $C$. We do not know $C$, but we want to output a set that is as close to $C$ as possible. The property of being "close" to $C$ corresponds to a performance guarantee that we will formulate in different ways. We may want to output a set not much larger than $C$ that is guaranteed to completely contain it, or we may want a small set that is guaranteed to have substantial overlap with $C$. A simple instance of the task is depicted in Figure 1b, after the explicit labeling of the core nodes has been removed from Figure 1a.

Generically, planted problems arise when some hidden structure (like the vertex cover in our case) has been "planted" in a larger input, and the goal is to find the structure. Planted problems are typically

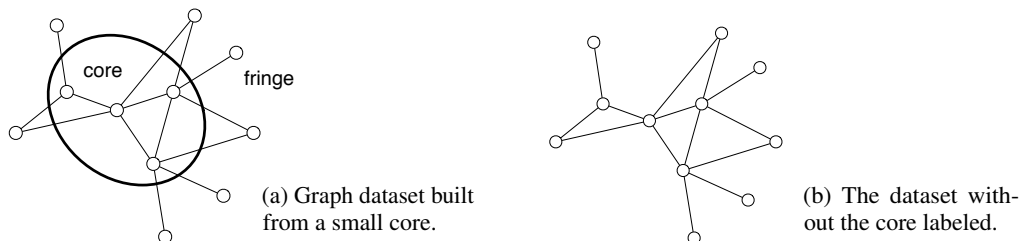

(a) Graph dataset built from a small core.

(b) The dataset without the core labeled.

Figure 1: (a) Graph datasets are often constructed by recording the interactions of a set of *core nodes*. The resulting data contains these core nodes together with a potentially much larger *fringe*, consisting of all other nodes that had an interaction with some member of the core. (b) Knowing which nodes are in the core is important for interpreting the dataset. However, in many cases, the graph is "found data" and this metadata is not available. This can arise from challenges in data provenance that lead to the loss of the metadata or when only partial information is available in contexts such as counter-surveillance. We study how accurately we can recover the core, despite limited information about how the dataset was constructed. Algorithmically, this leads to a *planted vertex cover* problem.

based on formal frameworks in which the input is generated by a highly structured probabilistic model. This is true in some of the most studied planted problems, such as the planted clique problem [5, 20, 25, 46] and the recovery problem for stochastic block models [1, 2, 3, 7, 19, 48].

It might seem inevitable that planted problems should require such probabilistic assumptions—how else could an algorithm guess which part of the graph corresponds to the planted structure, if there are no assumptions on what the "non-planted" part of the graph looks like? But the vertex cover problem turns out to be different, and it is possible to solve what, surprisingly, can be described as a "worst-case" planted problem. With extremely limited assumptions, we can provide provable guarantees for approximately recovering unknown vertex covers. Specifically, we make only two assumptions (both necessary in some form, though relaxable): (i) the planted vertex cover is inclusion-wise minimal and (ii) its size is upper-bounded by a known quantity $k$. We think of $k$ as small compared to the total number of nodes, which is consistent with many datasets. Among other results, we use fixed-parameter tractability to show that there is an algorithm operating on arbitrary graphs that can output a set of $f(k)$ nodes (independent of the size of the graph) guaranteed to contain the planted vertex cover. We obtain further bounds with additional structural assumptions.

We pair our theoretical guarantees with experiments showing the efficacy of these methods on real-world networks with core-fringe structure. Using our theory, we develop a natural heuristic based on constructing a union of minimal vertex covers from random initializations. Our simple algorithm provides superior recovery performance and running time compared to a number of competitive baselines. Among these, we show improvements over a line of well-developed heuristics for detecting *core-periphery structure* [15, 31, 53, 63]—a sociological notion in which a network has a dense core and a sparser periphery, generally for reasons of differential status rather than measurement effects.

## 2 Theoretical methodology for partial recovery or tight containment

We first formalize the problem. Suppose there is a large universe of nodes $U$ that interact via communication, friendship, or another mechanism. We choose a *core* subset $C \subseteq U$ and measure all pairwise interactions that involve at least one node in $C$. We represent our measurement by a graph $G = (V, E)$, where $V \subseteq U$ is all nodes that belong to $C$ or interact with at least one node in $C$, and $E$ is the set of all such interactions. The nodes in $V - C$ are called the *fringe* of the graph. We ignore directionality and self-loops, so $G$ is a simple, undirected graph. Note that $C$ is a vertex cover of $G$.

### 2.1 Finding a planted vertex cover

In the planted vertex cover problem, we observe $G$ and are tasked with finding $C$. Can we say anything non-trivial in answer to this question? Without other information, it could be that $C = V$, so we first assume that $|C| \leq k$, where we think of $k$ as small relative to $|V|$. With this extra information, we can ask if it is possible to obtain a small set that is guaranteed to contain $C$:

**Question 1.** *For some function $f$, can we find a set $D$ of size at most $f(k)$ (independent of the size of $V$) that is guaranteed to contain the planted vertex cover $C$?*

The answer to this question is "no." For example, let $k = 2$ and let $G$ be a star graph with $n > 3$ nodes $v_1, ..., v_n$ and edges $(v_1, v_i)$ for each $i > 1$. The two endpoints of any edge in the graph form a vertex cover of size $k = 2$, but $C$ could conceivably be any edge. Thus, under these constraints, the only set guaranteed to contain $C$ is the entire node set $V$.

In this negative example, once we put $v_1$ into a 2-node vertex cover, the other node can be arbitrary, since it is superfluous. This suggests that it would be more reasonable to ask about minimal vertex covers. Formally, $C$ is a *minimal vertex cover* if for all $v \in C$, the set $C - \{v\}$ is not a vertex cover. (In contrast, a *minimum* vertex cover is a minimal cover of minimum size.) Minimality is natural with respect to our motivating applications—it is reasonable to assume that the measured nodes are non-redundant, in the sense that omitting a node from $C$ would cause at least one edge to be lost from the measured communication pattern. Thus, we ask the following adaptation of Question 1:

**Question 2.** *If $C$ is a minimal planted vertex cover, can we find a set $D$ of size at most $f(k)$ (independent of $|V|$) that is guaranteed to contain $C$?*

Interestingly, the answer to this question is "yes" for arbitrary graphs. We derive this as a consequence of results from Damaschke and Molokov [17, 18] in the theory of fixed-parameter tractability:

**Lemma 1** ([17, 18]). *Consider a graph $G$ with a* minimum *vertex cover size $k^*$. Let $U(k)$ be the union of all* minimal *vertex covers of size at most $k$. Then*

    *(a) $|U(k)| \leq (k+1)^2/4 + k$ and is asymptotically tight [17, Theorem 3]*
    *(b) $|U(k)| \leq (k - k^* + 2)k^*$ and is tight [18, Theorem 12]*

There is an informative direct proof of part (a) that gives the $O(k^2)$ bound, using a *kernelization* technique from fixed-parameter tractability [12, 21]. The proof begins from the following observation:

**Observation 1.** *Any node with degree strictly greater than $|C|$ must be in $C$.*

The observation follows simply from the fact that if a node is omitted from $C$, then all of its neighbors must belong to $C$. Thus, if $S$ is the set of all nodes in $G$ with degree greater than $k$, then $S$ is contained in every vertex cover of size at most $k$. Hence Observation 1 implies that if $U(k)$ is non-empty, we must have $|S| \leq k$ and $S \subseteq U(k)$. Now $G - S$ is a graph with maximum degree $k$ and a vertex cover of size $\leq k$, so it has at most $O(k^2)$ edges. Next, let $T$ be the set of all nodes incident to at least one of these edges. Any node not in $S \cup T$ is isolated in $G - S$ and hence not part of any minimal vertex cover of size $\leq k$. Therefore $U(k) \subseteq S \cup T$, and so $|U(k)| = O(k^2)$. We will later use Observation 1 as motivation for a degree-ordering component to our planted vertex cover recovery algorithm.

The following theorem, giving a positive answer to Question 2, is thus a corollary of Lemma 1(a) obtained by setting $D = U(k)$, the union of all minimal vertex covers of size at most $k$:

**Theorem 1.** *If $C$ is a minimal planted vertex cover with $|C| \leq k$, then we can find a set $D$ of size $O(k^2)$ that is guaranteed to contain $C$.*

To see why $O(k^2)$ is tight, let $G$ consist consist of the disjoint union of $k/2$ stars each with $1 + k/2$ leaves. Any set consisting of the centers of all but one of the stars, and the leaves of the remaining star, is a minimal vertex cover of size $k$. Thus every node in $G$ could potentially belong to the planted vertex cover $C$, and so we must output the full node set $V$. Since $|V| = \Omega(k^2)$, the bound follows.

We can compute $U(k)$ in time exponential in $k$ but polynomial in the number of nodes $n$ for fixed $k$ [18], but these algorithms remain impractical for our datasets. However, the results motivate our algorithm in Section 3, which is based on taking the union of several minimal vertex covers.

## 2.2 Non-minimal vertex covers

A natural next question is whether we can say anything positive when the planted vertex cover $C$ is not minimal. In particular, if $C$ is not minimal, can we still ensure that some parts of it must be contained in $U(|C|)$, the union of all minimal vertex covers of size at most $|C|$? The following propositions show that if $u \in C$ links to a node $v$ either outside $C$ or deeply contained in $C$ (with $v$ and its neighbors all in $C$), then $u$ must belong to $U(|C|)$.

**Proposition 1.** *If $u \in C$ and there is an edge $(u, v)$ to a fringe node $v \notin C$, then $u \in U(|C|)$.*

*Proof.* Consider the following iterative procedure for "pruning" the set $C$. We repeatedly check whether there is a node $w$ such that $C - \{w\}$ is still a vertex cover. Anytime we encounter such a node $w$, we delete it from $C$. When this process terminates, we have a minimal vertex cover $C' \subseteq C$. Since $|C'| \leq |C|$, we must have $C' \subseteq U(|C|)$. But in this iterative process we cannot delete $u$, since $(u, v)$ is an edge and $v \notin C$. Thus $u \in C'$, and hence $u \in U(|C|)$. $\quad\square$

Next, let us say that a node $v$ belongs to the *interior* of the vertex cover $C$ if $v$ and all the neighbors of $v$ belong to $C$. We have the following result.

**Proposition 2.** *If $u \in C$ and there is an edge $(u, v)$ to a node $v$ in the interior of $C$, then $u \in U(|C|)$.*

*Proof.* Let $u$ and $v$ be nodes as described in the statement of the proposition. Since all of $v$'s neighbors are in $C$, it follows that $C_0 = C - \{v\}$ is a vertex cover. We now proceed as in the proof of Proposition 1. We iteratively delete nodes from $C_0$ as long as we can preserve the vertex cover property. When this process terminates, we have a minimal vertex cover $C' \subseteq C_0$, and since $|C'| \leq |C|$, we must have $C' \subseteq U(|C|)$. Now, $u$ could not have been deleted during this process, because $(u, v)$ is an edge and $v \notin C_0$. Thus $u \in C'$, and hence $u \in U(|C|)$. $\quad\square$

Even with these results, an arbitrarily small fraction of a non-minimal planted vertex cover $C$ may be in $U(|C|)$. Consider a star graph with center $u$ and $k + 1$ leaves, and let $C$ consist of $u$ and any $k - 1$ leaves. The set $\{u\}$ is the only minimal vertex cover of size $\leq k$, and hence $|U(|C|)|/|C| = 1/k$. In this example, only node $u$ satisfies the hypotheses of Proposition 1 or 2. However, we will see in Section 3 that in many real-world networks, most nodes in $C$ are indeed contained in $U(|C|)$ by satisfying one a condition in Proposition 1 or Proposition 2. The planted vertex cover recovery

algorithm that we develop in Section 3 uses the union of minimal vertex covers to identify nodes that are likely in the planted cover. Propositions 1 and 2 show that even if the planted cover is not minimal, we can still recover its nodes with such unions of minimal vertex covers.

The above example has the property that $C$ is much larger than the minimum vertex cover size $k^*$. We next consider the case in which $C$ may be non-minimal, but is within a constant factor of $k^*$. In this case, we show how to find small sets guaranteed to intersect a constant fraction of the nodes in $C$.

## 2.3 Maximal matching 2-approximation to minimum vertex cover and intersecting the core

A basic building block for our theory and algorithms is the classic maximal matching 2-approximation to minimum vertex cover (Algorithm 1, right). The greedy algorithm builds a maximal matching $M$ by processing each edge $e = (u, v)$ of the graph and adding $u$ and $v$ to $M$ if neither endpoint is already in $M$. Upon termination, $M$ is both a maximal matching and a vertex cover. It is maximal because if we could

---

**Algorithm 1:** Greedy maximal matching with 2-approximation for minimum vertex cover.

---

**Input:** Graph $G = (V, E)$
**Output:** Vertex cover $M$ with $|M| \leq 2k^*$
$M \leftarrow \varnothing$
**for** $e = (u, v) \in E$ **do**
$\quad \lfloor$ **if** $u, v \notin M$ **then** $M \leftarrow M \cup \{u, v\}$

---

add another edge $e$, then it would have been added when processed; and it is a vertex cover because if both endpoints of an edge $e$ are not in the matching, then $e$ would have been added to the matching when processed. Since any vertex cover must contain at least one endpoint from each edge in the matching, we have $k^* \geq |M|/2$, or $|M| \leq 2k^*$, where $k^*$ is the minimum vertex cover size of $G$. The output $M$ of Algorithm 1 may not be a *minimal* vertex cover. However, we can iteratively prune nodes from $M$ to make it minimal, which we do for our recovery algorithm described in Section 3. For the theory in this section, though, we assume no such pruning.

The following lemma shows that any vertex cover whose size is bounded by a multiplicative constant of $k^*$ must intersect the output of Algorithm 1 in a constant fraction of its nodes.

**Lemma 2.** *Let $B$ be any vertex cover of size $|B| \leq bk^*$ for some constant $b$. Then any set $M$ produced by Algorithm 1 satisfies $|M \cap B| \geq \frac{1}{2b}|B|$.*
*Proof.* The maximal matching $M$ is $h$ edges satisfying $h = |M|/2 \leq k^*$. Since $B$ is a vertex cover, it must contain at least one endpoint of each of edge in $M$. Hence, $|M \cap B| \geq h \geq k^*/2 \geq \frac{1}{2b}|B|$. □

A corollary is that if our planted cover $C$ is relatively small in the sense that it is close to the minimum vertex cover size, then Algorithm 1 must partially recover $C$. We write this as follows.

**Corollary 1.** *If the planted vertex cover $C$ has size $|C| \leq ck^*$, then Algorithm 1 produces a set $M$ of size $\leq 2k^*$ that intersects at least a $1/(2c)$ fraction of the nodes in $C$.*

The planted vertex cover recovery algorithm that we design in Section 3 guesses that nodes output by Algorithm 1 are part of the planted cover. Corollary 1 thus tells us that our method will indeed capture part of the planted cover.

An important property of Algorithm 1 that will be useful for our algorithm design later in the paper is that the algorithm's guarantees hold regardless of the order in which the edges are processed. Furthermore, two matchings produced by the algorithm using two different orderings of the edges must share a constant fraction of nodes, as formalized in the following corollary.

**Corollary 2.** *Any two sets $S_1$ and $S_2$ obtained from Algorithm 1 (with possible different orders in the processing of the edges) satisfy $|S_1 \cap S_2| \geq \frac{1}{4} \max(|S_1|, |S_2|)$.*

Our results thus far only assume that the graph contains a vertex cover of size at most $k$. Next, we show how assuming structure on the graph can yield stronger guarantees.

## 2.4 Improving results with known graph structure

We now strengthen our theoretical guarantees by assuming structure on $C$ and $G$. Specifically, we consider how to make use of random structure and bounds on the minimum vertex cover size $k^*$ obtained through computation with Algorithm 1 can strengthen our theoretical guarantees. These results are theoretical and do not affect the algorithms we develop in Section 3.

**Stochastic block model.** One possible structural assumption is that edges are generated independently at random. The stochastic block model (SBM) is a common generative model for this idealized

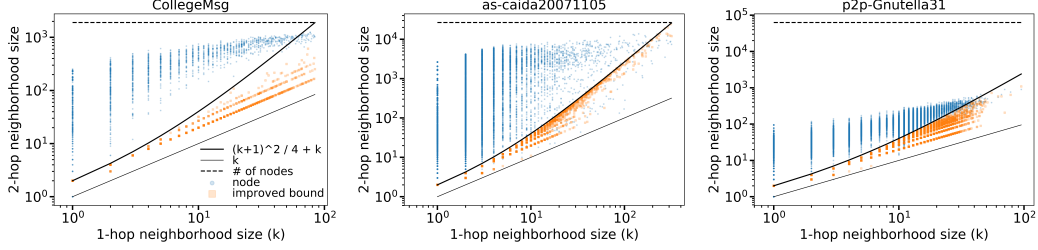

Figure 2: Improving Lemma 1(b) by bounding the minimum vertex cover size with the output of Algorithm 1. For each network dataset, we synthetically create a planted vertex cover $C$ for every 1-hop neighborhood, which covers the corresponding 2-hop neighborhood. Blue dots show relative sizes of the planted cover (1-hop neighborhood) and the subgraph it covers (the 2-hop neighborhood). The bound on $|U(|C|)|$ from Lemma 1(a) is the thick black line. Orange squares show improvements in the bound on $|U(|C|)|$ from Lemma 1(b) by bounding the minimum vertex cover size with Algorithm 1. The improved bounds appear linear instead of quadratic for CollegeMsg.

setting [30] that is the basis for a major class of planted problems [1]. Here we consider a 2-block SBM, where one block is the planted vertex cover $C$ and the other block is the fringe nodes $F$. The SBM provides a probability of an edge forming within a block and between blocks. For our purposes, $C$ is a vertex cover, so we assume that the probability of an edge between nodes in $F$ is 0. Denote the probability of an edge between nodes in $C$ as $p$ and the probability of an edge between a node in $C$ and a node in $F$ as $q$. We make no assumption on the relative values of $p$ and $q$.

The following result combines Lemma 1(b) with well-known lower bounds on independent set size in Erdős-Rényi graph (in the SBM, $C$ is an Erdős-Rényi graph with edge probability $p$).

**Lemma 3.** *With probability at least $1 - |C|^{-3 \ln n/(2p)}$, the union of minimal vertex covers of size at most $|C|$ contains at most $|C|(3 \ln |C|/p + 3)$ nodes.*

*Proof.* It is straightforward to show that the independence number $\alpha$ of $C$ is less than $(3 \ln |C|)/p + 1$ with probability at least $1 - |C|^{-3 \ln |C|/(2p)}$ [57]. The minimum vertex cover size of the first block is then $k^* = |C| - \alpha \geq |C| - (3 \ln |C|)/p - 1$ with the same probability. Plugging this bound on $k^*$ into Lemma 1(b) gives the result. $\square$

The following theorem further develops how the SBM provides substantial structure for our problem.

**Theorem 2.** *Let $C$ be a planted vertex cover in our SBM model, where $|C| = k$. Let $p$ and $q$ be constants, and let the number of nodes in the SBM be $ck$ for some constant $c \geq 1$. Then with high probability as a function of $k$, there is a set $D$ of size $O(k \log k)$ that is guaranteed to contain $C$.*

*Proof.* By Lemma 3, we know that $|U(k)|$ is $O(k \log k)$ with high probability. Now, for any node $v \in C$, the probability that it links to at least one node outside of $C$ is $1 - (1 - q)^{(c-1)k}$. Taking the union bound over all nodes in $C$ shows that with high probability in $k$, each node in $C$ has at least one edge to a node outside $C$. In this case, Proposition 1 implies that $C \subseteq U(k)$, so by computing $U(k)$, we contain $C$ with high probability. $\square$

**Bounds on the minimum vertex cover size.** It may be impractical to compute the *minimum* vertex cover size $k^*$, but Lemma 1(b) may still be used if we can bound $k^*$ from above and below. Specifically, given lower and upper bounds $l$ and $u$ on $k^*$, $|U(k)| \leq (k - l + 2)u$. A cheap way to find such bounds is to use Algorithm 1 several times with different edge orderings (and subsequent pruning) to get better lower and upper bounds on $k^*$. We evaluated this methodology on three datasets: (i) a network of messages sent on a college's online social network (CollegeMsg; [51]), (ii) an autonomous systems graph (as-caida20071105; [41]), and (iii) a snapshot of the Gnutella P2P file sharing network (p2p-Gnutella31; [45]). For each network, we construct many planted covers by considering a 1-hop neighborhood of a node $u$ covering the 2-hop neighborhood of $u$ (we then remove $u$ from these sets). We used 20 random orderings of the edges with Algorithm 1 to get the best bound on $|U(k)|$. Figure 2 summarizes the results. In most cases, the approximations substantially improve the bound. With CollegeMsg, the bounds appear approximately linear in the cover size.

## 3 Recovery performance on datasets with "real" planted vertex cores

Next, we study recovery of planted vertex covers in datasets with conceivable but known core-fringe structure arising from the measurement processes described in the introduction. We use 5 datasets:

```
1  using SparseArrays, Random
2  function UMVC(A::SparseMatrixCSC{Int64,Int64},
3                  ncovers::Int64=300)
4    edges = filter(e->e[1] < e[2],
5                  collect(zip(findnz(A)[1:2]...)))
6    umvc = zeros(Int64,size(A,1))
7    for _ in 1:ncovers
8      # Run 2-approximation with random edge ordering
9      vc = zeros(Int64, size(A,1))
10     for (i, j) in shuffle(edges)
11       if vc[[i,j]] == [0,0]; vc[[i,j]] .= 1; end
12     end
13     # Reduce to a minimal cover
14     while true
15       vc_size = sum(vc)
16       for c in shuffle(findall(vc .== 1))
17         nbrs = findnz(A[:,c])[1]
18         if sum(vc[nbrs]) == length(nbrs); vc[c] = 0; end
19       end
20       if sum(vc) == vc_size; break; end
21     end
22     umvc[findall(vc .== 1)] .= 1
23   end
24   degs = vec(sum(A, dims=1)) # node degrees
25   return sortperm(collect(zip(umvc, degs)), rev=true)
26 end
```

Figure 3: Complete implementation of our union of minimal vertex covers (UMVC) algorithm in 26 lines of Julia code. The algorithm repeatedly runs the standard maximal matching 2-approximation algorithm for minimum vertex cover (Algorithm 1; in lines 8–12) and reduces each cover to a minimal one (lines 13–21). The union of covers is ranked first in the ordering, sorted by degree; the remaining nodes are then sorted by degree (line 25). Code on the left is available at https://gist.github.com/arbenson/27c6d9ef2871a31cbdbba33239ea60d0.

*(1) email-Enron* [35] is an email communication network, where the core is the set of email addresses whose inboxes were released via a regulatory investigation; *(2) email-W3C* [14, 50, 61] is derived from crawled W3C email list threads, where the core is the set of nodes with a w3.org domain in the email address; *(3) email-Eu* [42, 62] consists of emails involving members of a European research institution, where the core nodes are the institution's members. *(4) call-Reality and (5) text-Reality* [22] come from phone calls and text messages involving a set of students and faculty at MIT participating in the reality mining project. The study participants constitute the core, and an edge connects two phone numbers if a call or message was made between them. Each dataset has timestamped edges, and we will evaluate how well we can recover the core as the networks evolve.

Table 1 provides summary statistics of the datasets. We include the minimum vertex cover size, which lets us evaluate Lemma 1(b). We also computed the fraction of nodes that are guaranteed to be in $U(|C|)$ by Propositions 1 and 2 and find that 82%–99% of the nodes fit these guarantees.

We next study recovery of the planted vertex cover, i.e., the core $C$. The methods we use provide a node ordering, often through a score function. We then evaluate recovery using precision at core size (the fraction of the top-$|C|$ ordered nodes that are in $C$) and area under the precision recall curve.

**Proposed algorithm: union of minimal vertex covers (UMVC).** Our proposed algorithm, which we call the union of minimal covers (UMVC), repeatedly finds minimal vertex covers and takes their union. The nodes in this union are ordered by degree first and the nodes not appearing in any minimal cover are ordered by degree after. The minimal covers are constructed by first finding a 2-approximate solution to the *minimum* vertex cover problem using Algorithm 1 (which takes linear time in the size of the data) and then pruning the resulting cover to be minimal. We randomly order the edges for processing in order to capture different minimal covers (we use 300 covers in our experiments). The algorithm is incredibly simple. Figure 3 shows a complete implementation in just 26 lines of Julia code. We use 300 covers as this keeps the running time to about a minute on the largest dataset. However, a smaller number is needed for the same performance on several datasets. In practice, a larger number of covers requires more computation but could capture more nodes in the planted cover. At the same time, a larger number of covers could lead to more false positives.

Table 1: Summary statistics of datasets: number of nodes ($n$), number of edges ($m$), time spanned, planted vertex cover size ($|C|$), minimum vertex cover size ($k^*$), bounds from Lemma 1 as a fraction of the total number of nodes (trivially capped at 1), and fraction of nodes in $C$ connected to a node $v$ (i) not in $C$ or (ii) in the interior of $C$ (all neighbors are in $C$). Nodes in the last two cases are guaranteed to be in the union of minimal vertex covers of size at most $|C|$ by Propositions 1 and 2.

| Dataset | n | m | days spanned | $|C|$ | $k^*$ | Bnd. a | Bnd. b | frac. Prop. 1 | frac. Prop. 2 |
|---|---|---|---|---|---|---|---|---|---|
| email-Enron | 18.6k | 43.2k | 1.50k | 146 | 146 | 0.30 | 0.02 | 0.99 | 0.00 |
| email-W3C | 20.1k | 31.9k | 7.52k | 1.99k | 1.11k | 1.00 | 1.00 | 0.76 | 0.06 |
| email-Eu | 202k | 320k | 804 | 1.22k | 1.18k | 1.00 | 0.26 | 0.99 | 0.00 |
| call-Reality | 9.02k | 10.6k | 543 | 90 | 82 | 0.24 | 0.09 | 0.90 | 0.01 |
| text-Reality | 1.18k | 1.95k | 478 | 84 | 80 | 1.00 | 0.41 | 0.88 | 0.00 |

Importantly, UMVC *makes no assumption on the size or minimality of the planted cover C*. Instead, we are motivated by our theory in Section 2 in several ways. First, we expect that most of $C$ will lie in the union of all minimal vertex covers of size at most $|C|$ by Propositions 1 and 2 and Theorem 1. The degree ordering is motivated by Observation 1, which says that nodes of sufficiently large degree must be in $C$. Second, even though we are pruning the maximal matchings to be minimal vertex covers, Corollary 1 provides motivation that the matchings should be intersecting $C$. If only a constant number of nodes are pruned when making the matching a minimal cover, then the overlap is still a constant fraction of $C$. Third, Corollary 2 says that we shouldn't expect the union to grow too fast.

**Other algorithms.** We compare UMVC against 5 other methods. First, we use a degree ordering of nodes, which captures the fact that the nodes outside of $C$ cannot link to each other and that $|C|$ is much smaller than the total number of vertices. This heuristic is a common baseline for core-periphery identification [53] and is theoretically justified in certain stochastic block models of core-periphery structure [63]. Second, we order nodes by betweenness centrality [26], the idea being that nodes in the core must appear in shortest paths between fringe nodes. Third, we order nodes by Path-Core (PC) scores [16], which have been used to identify core-periphery structure in networks [40]. Fourth, we use a metric from Borgatti and Everett (BE) which scores nodes in a way to capture core-periphery structure [8, 53]. Fifth, we use a belief propagation (BP) method for block recovery in stochastic block models of core-periphery structure [63].

**Recovery performance.** We divide the temporal edges of each dataset into 10-day increments and construct an undirected, unweighted, simple graph for the first $10r$ days of activity, $r = 1, 2, \dots, \lfloor T/10 \rfloor$, where $T$ is the number of days spanned by the dataset. Given the ordering of nodes from an algorithm, we evaluate recovery performance by the precision at core size (P@CS; Figure 4, left) and area under the precision recall curve (AUPRC; Figure 4, right). We also provide upper bounds on performance. This is the fraction of core nodes that are non-isolated for P@CS and the AUPRC of a node ordering that places all non-isolated core nodes first and then the remaining nodes randomly.

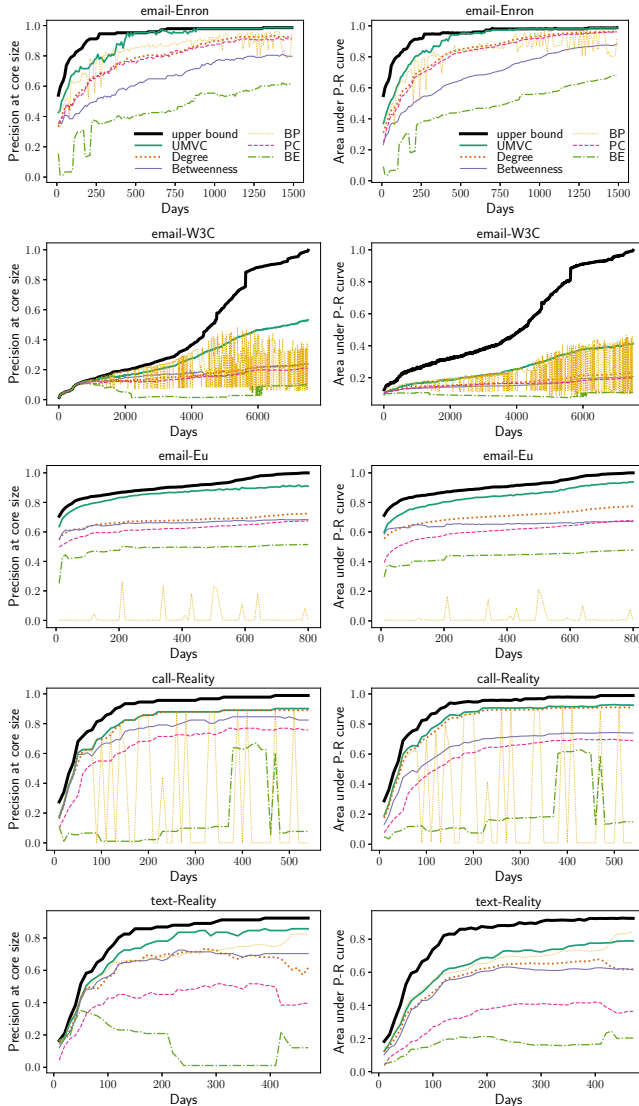

Figure 4: Core recovery performance on real-world datasets using our union of minimal vertex covers algorithm (UMVC), degree ordering, betweenness centrality [26]), belief propagation (BP; [63]), Borgatti-Everett scores (BE; [8]), and Path-Core scores (PC; [16]). Each algorithm orders the nodes, and we measure performance with precision at core size (left) and area under the precision recall curve (right) over every 10 days of real time. UMVC performs well on all datasets. BP is sometimes competitive but is susceptible to poor local minima, giving erratic performance. UMVC is also much faster than betweenness, BP, and PC (Table 2).

Table 2: Running time of algorithms on our datasets (here, we use the entire dataset, instead of evaluating over time, as in Figure 4). Our proposed union of minimal vertex covers algorithm (UMVC) is fast and provides the best performance on several real-world datasets (Figure 4).

| Dataset | UMVC | degree | betweenness [26] | PC [16] | BE [8, 13] | BP [63] |
|---|---|---|---|---|---|---|
| email-W3C | 6.5 secs | < 0.01 secs | 2.8 mins | 1.1 hours | 0.1 secs | 1.0 mins |
| email-Enron | 8.4 secs | < 0.01 secs | 2.5 mins | 1.8 hours | 0.1 secs | 20.2 mins |
| email-Eu | 1.2 mins | < 0.01 secs | 11.8 hours | > 3 days | 0.9 secs | 15.0 mins |
| call-Reality | 2.2 secs | < 0.01 secs | 27.9 secs | 6.1 mins | 1.8 secs | 4.3 secs |
| text-Reality | 0.5 secs | < 0.01 secs | 0.8 secs | 11.4 secs | < 0.1 secs | 6.8 secs |

Across all datasets, our UMVC algorithm out-performs the degree, betweenness, PC, and BE baselines at nearly all points in time. BP sometimes exhibits slightly better performance but suffers from erratic performance over time due to landing in local minima; see, e.g., results for email-W3C (Figure 4, row 2). In some cases, BP can hardly pick up any signal, such as in email-Eu (Figure 4, row 3). On the email-Eu dataset, UMVC vastly out-performs all other baselines. On email-Enron, UMVC achieves perfect recovery after 900 days of activity.

A key reason for UMVC's success is its use of the vertex cover structure. Other algorithms do not detect low-degree nodes that might look like traditional "periphery" nodes while residing in the core. Core-periphery detection algorithms in network science have traditionally relied on SBM benchmarks or heuristic evaluation [53, 63]. We have already shown how the SBM induces substantial structure for our problem, so the SBM is not a great benchmark for further analysis. Here, we used some notion of ground truth labels on which to evaluate the algorithms and exploited the vertex cover structure to get good performance.

**Timing performance.** Table 2 shows the time to run the algorithms on the entire dataset. We used 300 vertex covers for UMVC, which was implemented in Julia (Figure 3). Tuning the number of covers provides a trade off between run-time performance and (potentially) recovery performance. The degree ordering was also implemented in Julia, and betweenness centrality was computed with the `LightGraphs.jl` Julia package's implementation of Brandes' algorithm [9, 10]. Path-Core scoring (PC) was implemented with Python's `NetworkX`, and the Belief Propagation (BP) algorithm was implemented in C++. We note that our goal is to demonstrate the approximate computation times, rather than to compare the most high-performance implementations. UMVC takes a few seconds on the email-W3C, email-Enron, call-Reality, and text-Reality datasets, and about one minute for the email-Eu dataset. This is an order of magnitude faster than BP, and several orders of magnitude faster than betweenness and PC. There are fast approximation algorithms for betweenness [6, 27], but the weak performance of exact betweenness did not justify exploring these approaches.

# 4 Discussion

Many network datasets are partial measurements and are often found in some way that destroys the record of how the measurements were made. Here, we examined the case where edges are collected by observing interactions involving some core set of nodes, and the identity of the core nodes is lost. Recovering the core nodes is then a planted vertex recovery problem. We developed theory for this problem, which we used to devise a simple algorithm that recovers the core with high efficacy in several real-world datasets. We assumed that our graphs were simple and undirected, but richer structure such as edge directions and timestamps could be incorporated in future research.

The network science community has tools for core-periphery detection, but evaluation is typically limited to recovery of synthetic models or ad-hoc score functions. Our work is also the first effort to evaluate the recovery of core-periphery-like (i.e., core-fringe) network structure through the lens of machine learning with "ground truth" labels on the nodes. We hope that this provides a valuable testbed for evaluating algorithms that reveal core-periphery or core-fringe structure, although we should not take evaluation on ground truth labels as absolute [52].

Code and data accompanying the experiments in this paper are available at
https://github.com/arbenson/FGDnPVC.

**Acknowledgments**

We thank Jure Leskovec for providing access to the email-Eu data; Mason Porter and Sang Hoon Lee for providing the Path-Core code; and Travis Martin and Thomas Zhang for providing the belief propagation code. This research was supported in part by a Simons Investigator Award, NSF TRIPODS Award #1740822, and NSF Award DMS-1830274.

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
