[Reviews · NeurIPS 2018]

Reviewer 1



Summary ------- In this paper, the authors study the problem of recovering a core set of nodes in a graph given the interactions between nodes of the graph. There are cases, where edges are collected by observing all interactions involving some core set of nodes and the identity of the core nodes is lost. This can happen in real scenarios, because (1) they are lost due to difficulties in data provenance, and (2) the network consists of “found data” obtained in settings such as counter-surveillance. Recovering the core nodes is a planted vertex cover problem. The proposed method gets as input a simple undirected graph (core set nodes + fringe nodes - nodes outside core set with interactions with nodes in core set) and the output is the core set (a set of nodes in the core set). The method is based on the theory of fixed-parameter tractability and algorithms for recovering the core. The authors have experimented with various real world datasets (with ground truth) and have compared with some baselines. The results show that the method proposed outperforms the comparison, it is faster (in runtime) and very simple to implement. Strong Points ------------- - With limited assumptions the paper provides provable guarantees for approximately recovering unknown vertex covers. - The paper contains well-defined theoretical analysis. - Experimentation and evaluation on a variety of datasets and well-known baseline comparison methods. Weak Points ----------- - The writing of the paper can be improved. Section 1 contains long sentences and as a result the reader might lose the focus that each sentence is aiming at. - The flow of the paper and more specifically Section 2 can be improved. There is an enumeration of theorems, lemmas, propositions, corollaries and only in Section 3 (where the algorithm is revealed) it becomes clear to the reader where all these lead to. Overall -------- I liked the paper a lot. The problem is related to NIPS crowd, it is well supported by real world applications (also with references) and the authors provide a well-defined algorithm with provable guarantees. It is very interesting that the implementation to solve this problem is only 30 lines long and simple to code. *** After reading author's feedback *** I am satisfied with the answers to all reviewers, especially to the reasonable comments by R3.

Reviewer 2



In this paper, the authors are formally defining the problem of core recovery of graphs through the lens of vertex cover. Specifically, the paper makes a connection with real-world applications through the way that most datasets are crawled; there is a set of `core` nodes that are flagged to be collected, and a set of `fringe` nodes that are in the dataset because of their interactions with the core. In the case where the `core` and `fringe` labels are hidden, the authors aim to provide theoretically-sound algorithms for recovering them. In the network science field, recovering the core of a network is a fairly common task. It helps the analyst gather intuition about which parts of the dataset are crucial and `important`. Oftentimes, the analyst wants to avoid making claims for the fringe nodes, especially since the collection task is not focusing on them. As the authors note, there have been many, mostly heuristic, methods that are trying to approach this problem indirectly. Here, the authors propose an algorithm that is directly dealing with the core recovery. Additionally, the proposed algorithm comes with theoretical guarantees, which make it more elegant and more powerful than adhoc heuristics. Apart from the elegant formulation and the careful theoretical analysis that the authors include in this work, there is also an adequate practical analysis of the proposed algorithm on real data. The comparison in the experimental section is made against a large number of well-known heuristics. The results indicate the power of the algorithm, and its advantage over the baseline. While the paper is written carefully and detailed, parts of the writing can be improved towards an smoother flow. Section 2, which is the core part of this paper, is very dense and contains too much information without a strong storyline. In particular, it is not easy for the reader to guess how each of the theorems, lemmas and propositions will inform the design of the algorithm. In other words, the authors try to enumerate all possible theoretical results and then summarize them into Algorithm UMVC. My suggestion would be that the authors try to hint the algorithm design early on, so that the reader can more easily follow the theoretical analysis; it is always better to understand ahead of time where each theorem will become useful. Specifically, the authors should focus their rewriting in Section 2.2, which is unclear what purpose it serves until Section 3. Another place that needs more attention is Figure 2 and the text that accompanies it. It is not clear what its purpose is, its legend is not very informative. Additionally, is the "Stochastic Block Model" paragraph of Section 2.4 used anywhere in the algorithm, or is it just an observation? Finally, it would improve the clarity if the theoretical guarantees of the algorithm together with its runtime complexity are summarized in a Lemma, so that the reader can easily refer to this information without having to scan the whole paper. Overall, this paper formulates a well-studied problem in an elegant way, and proposes a theoretically-sound algorithm for this problem. In fact, this algorithm will have significant impact in many real-world data analytics tasks. The clarity of the writing, although not bad, has room for improvement. The authors, although targeting for a more theoretical type of writing, include an adequate experimental section ---- After reading the authors' response to the reviews, I am satisfied with their answers to the very reasonable points raised by Reviewer 3. I am also happy that my own comments proved to be helpful as well.

Reviewer 3



This paper considers graphs that are partitioned into two separate components called a "core" (C) and a "fringe" (F), where all edges have at least one endpoint in C. The partitioning is supposed to be unknown, and the goal of the paper is to come up with an algorithm that recovers it. Strong aspects of the paper are: - The paper first performs a theoretical analysis, deriving various conditions for the set C and upper and lower bounds for its size under various assumptions. Based on the theory, it then proposes an algorithm for recovering C, and shows that this algorithm outperforms various baselines. - The theoretical analysis is sound and intuitive. The theory nicely and naturally leads to an algorithm that is indeed surprisingly simple and performs well empirically. - The paper is very well written, the motivation is clearly explained, the theoretical results are clearly structured and nicely followable. Most of the theoretical results are relatively straightforward with a few exceptions (Lemma 3 and Theorem 2). - The considered problem is a special case of "core-periphery" structure identification in networks (e.g. Zhang et al, 2015, Phys Review E; their ref 61). The contribution rests on the assumption that no links at all exist between nodes in the periphery. This assumption leads to an elegant algorithm that can convincingly outperform others when the assumption holds. The techniques are potentially of interest for other researchers in the networks field. I have three main criticisms: - The restriction to no edges in F seems very restrictive and I am not convinced by the given examples that this is a problem that would be of interest in the real world. In the used datasets, the authors dropped the available labels of C and F. In the motivation given in the beginning, they say that in some applications these labels are lost, but can they point to any real datasets that have this property or even better, analyze one and present the results? - It is perhaps not too surprising that the proposed algorithm outperforms baseline given that the baseline methods were developed for a more general scenario and thus do not take advantage of the absence of edges in F. It would have been interesting to see how the proposed algorithm would fare on more general community structure detection problems such as the blog link data shown in Zhang et al, 2015, Phys Review E. Would the algorithm still provide a useful approximation in this case, or would it break down completely? - Finally, I wonder if NIPS is the appropriate avenue for this research area. Networks has traditionally not been a topic with a very strong presence at NIPS as far as I am aware, and more appropriate venues that would might to mind would be KDD, ICDM. A possible sign of mismatch is that the paper does not cite any paper from NIPS or any other ML/AI venue. Perhaps the authors can explain in their rebuttal why they chose this specific conference? Minor points: - I would have liked to see more justification for the choice of the number of clusters to be generated (fixed here at 300). Is this important for performance? Under which circumstances would I need to set this higher or lower? - It was hard to compare the numbers in Table 2 due to different units, can this be standardized to one unit or alternatively, can at least the fastest algorithm be highlighted in each row? = After author response = The authors have presented some new data that addresses my second criticism and since no other reviewer had any doubts about suitability for NIPS, I retract that criticism as well. The authors cite personal communications as evidence for the importance of the considered setting, which still does not convince me entirely. However, I am satisfied with the response overall and will revise my score up by 1.